# Maresin1 Alleviates Ischemia Reperfusion Injury After Lung Transplantation by Inhibiting Ferroptosis via the PKA-Hippo-YAP Signaling Pathway

**DOI:** 10.3390/biomedicines13071594

**Published:** 2025-06-30

**Authors:** Peng Deng, You Wu, Li Wan, Xiangfu Sun, Quanchao Sun

**Affiliations:** 1Division of Cardiovascular Surgery, Tongji Hospital, Tongji Medical College, Huazhong University of Science and Technology, Wuhan 430030, China; pengdeng@hust.edu.cn; 2Department of Gastrointestinal Surgery, Union Hospital, Tongji Medical College, Huazhong University of Science and Technology, Wuhan 430022, China; wuyouxiehe@163.com; 3Department of Thoracic Surgery, Union Hospital, Tongji Medical College, Huazhong University of Science and Technology, Wuhan 430022, China; wanli7582@163.com (L.W.); sunxiangfu123@outlook.com (X.S.)

**Keywords:** Maresin1, lung ischemia reperfusion injury, ferroptosis, signaling pathway, lung transplantation

## Abstract

**Background**: Lung ischemia reperfusion injury (LIRI) is a severe complication after lung transplantation (LT). Ferroptosis contributes to the pathogenesis of LIRI. Maresin1 (MaR1) is an endogenous pro-resolving lipid mediator that exerts protective effects against multiorgan diseases. However, the role and mechanism of MaR1 in the ferroptosis of LIRI after LT need to be further investigated. **Methods**: A mouse LT model and a pulmonary vascular endothelial cell line after hypoxia reoxygenation (H/R) culture were established in our study. Histological morphology and inflammatory cytokine levels predicted the severity of LIRI. Cell viability and cell injury were determined by CCK-8 and LDH assays. Ferroptosis biomarkers, including Fe^2+^, MDA, 4-HNE, and GSH, were assessed by relevant assay kits. Transferrin receptor (TFRC) and Acyl-CoA Synthetase Long Chain Family Member 4 (ACSL4) protein levels were examined by western blotting. In vitro, lipid peroxide levels were detected by DCFH-DA staining and flow cytometry analysis. The ultrastructure of mitochondria was imaged using transmission electron microscopy. Furthermore, the potential mechanism by which MaR1 regulates ferroptosis was explored and verified with signaling pathway inhibitors using Western blotting. **Results**: MaR1 protected mice from LIRI after LTx, which was reversed by the ferroptosis agonist Sorafenib in vivo. MaR1 administration decreased Fe^2+^, MDA, 4-HNE, TFRC, and ACSL4 contents, increased GSH levels, and ameliorated mitochondrial ultrastructural injury after LTx. In vitro, Sorafenib resulted in lower cell viability and worsened cell injury and enhanced the hallmarks of ferroptosis after H/R culture, which was rescued by MaR1 treatment. Mechanistically, the protein kinase A and YAP inhibitors partly blocked the effects of MaR1 on ferroptosis inhibition and LIRI protection. **Conclusions**: This study revealed that MaR1 alleviates LIRI and represses ischemia reperfusion-induced ferroptosis via the PKA-Hippo-YAP signaling pathway, which may offer a promising theoretical basis for the clinical application of organ protection after LTx.

## 1. Introduction

Lung transplantation (LTx) represents a pivotal therapeutic strategy for patients suffering from multiple end-stage pulmonary diseases; however, the survival rate of LTx is often hindered by the intricate pathophysiology of ischemia reperfusion injury [1,2]. Lung ischemia reperfusion injury (LIRI) occurs when blood flow is restored to the lung after a period of ischemia, leading to acute respiratory distress syndrome after LTx with impaired graft function and high mortality [3,4]. Furthermore, LIRI can also occur in a variety of clinical settings, such as pulmonary sleeve resection, pulmonary thromboembolism, cardiopulmonary bypass, aortic aneurysm repair, and tourniquet application for orthopedic operations. The pathophysiological process of LIRI includes a spectrum of events, ranging from oxidative stress, coagulation dysfunction, inflammation, endothelial damage, and cell death [5,6,7]. The mechanism of LIRI is still not fully clarified. Thus, exploring effective strategies to prevent LIRI is of great importance for improving the outcome of LTx.

Ferroptosis is a novel mode of programmed cell death that is characterized by iron-dependent peroxidation of lipid membranes [8,9]. Notably, recent studies indicated that abnormally activated ferroptosis triggers excessive inflammation to aggravate injury, which plays an important role in the pathogenesis of multiorgan ischemia reperfusion injury [10,11]. Inhibiting ferroptosis and ACSL4 can alleviate ferroptotic damage by reducing lipid peroxidation in LIRI [12,13]. The intricate interplay between ferroptosis and IRI unfolds as a complex process of molecular events, involving reactive oxygen species (ROS) accumulation, lipid peroxidation, and dysregulated iron metabolism [14,15,16]. Hence, targeting ferroptosis is a promising strategy for preventing LIRI.

Maresin1 (MaR1), a specialized pro-resolving lipid mediator derived from omega-3 polyunsaturated fatty acids, emerges as a key player in the inflammation resolution and tissue repair [17,18]. It is believed to attenuate the inflammatory factors, oxidative stress, and macrophage polarization triggered by IRI and dampen coagulation, neutrophil infiltration, and cell injury [19,20,21,22]. However, whether MaR1 can inhibit ferroptosis during LIRI after LTx remains unclear.

The Hippo signaling pathway, including MST1/2 kinase, LATS1/2 kinase, and the transcriptional co-activators YAP/TAZ, plays an important role in regulating various biological processes, such as cell proliferation, tissue regeneration, cell death, and mechanical sensing [23]. The activated LATS1/2 phosphorylates YAP and TAZ, leading to their cytoplasmic localization and subsequent low gene expression [24]. On the other hand, when the Hippo signaling pathway is inactivated, YAP and TAZ translocate to the nucleus, where they interact with transcription factors such as TEAD (TEA/ATTS domain) family members to promote gene overexpression [25,26]. A recent study demonstrated that TFRC and ACSL4 expression are regulated by the Hippo signaling pathway in ferroptosis [27].

In this study, we aimed to explore the role of MaR1 in the regulation of ferroptosis and the potential mechanism in LIRI, which mainly focused on the Hippo signaling pathway. To test our hypothesis, we performed the following experiments: (1) we established a LIRI model in vivo and a hypoxia reoxygenation (H/R) culture system in vitro and treated the recipient with MaR1 or vehicles in accordance with the group design; (2) we examined the lung injury, cell viability, and the hallmarks of ferroptosis in vivo and in vitro; (3) we evaluated the PKA-Hippo-YAP signaling pathway alterations and manipulated it with pharmacological inhibitors to verify the underlying mechanism of MaR1 on ferroptosis and lung injury.

## 2. Materials and Methods

### 2.1. Animals

Specific pathogen-free male 8–10-week-old C57BL/6 mice, weighing 30–35 g, were purchased from Shulaibao Co., Ltd. (Wuhan, China). All animals were provided with standard laboratory food and water in a temperature-controlled environment with a 12:12 h light/dark cycle (Tongji Medical College, Huazhong University of Science and Technology, Wuhan). Animal experimental procedures were established in accordance with the Guideline for Use and Care of Laboratory Animals (NIH publication, eighth edition) and approved by the Animal Care and Use Committee of Tongji Medical College of Huazhong University of Science and Technology.

### 2.2. Experiment Protocol

The LIRI model was induced by orthotopic lung transplantation in mice [28,29]. Briefly, donors and recipients were anesthetized with pentobarbital sodium (50 mg/kg) administered by intraperitoneal injection prior to surgery. For the donor, the pulmonary vein, artery, and bronchial cuff were made with 22-gauge, 24-gauge, and 20-gauge intravenous catheters, respectively. The distal ends of them were passed through the cuffs and then everted over and firmly fastened with a circumferential 10−0 silk suture. For the recipient, thoracotomy was performed through the left fifth intercostal space, and the left lung hilum was carefully isolated and occluded with a slip knot. Then, a tiny incision of approximately one-third of the circumference of the vessel or bronchus was made for the cuff implantation. After 3 h of cold ischemia, the cuffs of the graft lung were placed into the bronchus, pulmonary artery, and vein of the recipient and fixed with 10−0 silk sutures. The hilar clamp was removed, and the lungs were collected after 3 h of reperfusion. The mice in the sham group undergo a thoracotomy operation without LTx. Standardized surgical protocols were applied to minimize mechanical trauma (e.g., uniform anastomosis techniques, avoidance of excessive traction). Hypothermia was mitigated by maintaining systemic normothermia (36–37 °C) via forced-air warming devices, with continuous core temperature monitoring. MaR1 (1.0 ng) was freshly dissolved in 0.1 mL saline and intravenously injected 30 min before reperfusion, while the control and LIRI groups received an equal volume of normal saline. Sorafenib was given by oral gavage at a dose of 30 mg/kg to the recipient 24 h before LTx or 10 µM in H/R culture [30,31]. This schedule was selected on the basis of the long half-life of sorafenib [32]. To inhibit PKA signaling, KT5720, a selective PKA inhibitor, was applied. For YAP/TAZ inhibition, verteporfin, a small-molecule disruptor of YAP-TEAD interactions, was used in our experiment. KT5720 was administered to mice via intraperitoneal injection (10 mg/kg) 1 h prior to the MaR1 or 5 µM in H/R culture [33,34]. Verteporfin was administered via intraperitoneal (6 mg/kg) injection 1 h prior to the MaR1 or 5 µM in H/R culture [35,36].

### 2.3. Cell Culture

Primary Human Pulmonary Microvascular Endothelial Cells (HPMEC) were cultivated on 24-well plates for 48 h. For the sham group, cells were cultured under normoxic conditions (21% O_2_, 5% CO_2_, 37 °C) in complete growth medium. The H/R group was performed as follows. Serum-free/glucose-free endothelial cell medium was replaced after washing with PBS three times, then cells were transferred to a humidified, sealed hypoxia chamber at 37 °C to establish hypoxic conditions. After 3 h of hypoxia, cells were washed three times with PBS and cultivated for 3 h in normal endothelial growth medium at 37 °C in a 5% CO_2_ humidified incubator (reoxygenation condition) [37].

### 2.4. Reagents

Maresin1 (7R,14S-dihydroxy-4Z,8E,10E,12Z,16Z,19Z-docosahexaenoic acid, No. 10878) and Verteporfin (No. 17334) were purchased from Cayman (Ann Arbor, MI, USA). Sorafenib (No. SRP0702) and KT 5720 (No. 420323) were obtained from Sigma-Aldrich (Merck, Darmstadt, Germany). Anti-p-PKA (ab59218)/PKA (ab216572) antibody, anti-p-LATS1/2 (ab305029)/LATS1/2 (ab234820) antibody, and anti-p-YAP (ab283327)/YAP (ab52771) antibody were purchased from Abcam (Cambridge, UK). Anti-TFRC antibody (No. 14-0719-82) and anti-ACSL4 antibody (No. PA5-27137) were obtained from ThermoFisher Scientific (Waltham, MA, USA).

### 2.5. Histopathological Pathologic Analysis

Graft lung samples were fixed with 4% paraformaldehyde at room temperature for 24 h, embedded in paraffin, stained with hematoxylin and eosin (H&E), and finally imaged using a microscope. The lung injury scores were determined based on the histopathological scoring system by two observers blinded to the interventions for each group. This scoring system assesses the following features: ① Alveolar congestion: graded based on the extent of congestion visible in alveolar spaces. ② Hemorrhage: assessed by the presence and severity of bleeding within alveoli and interstitial spaces. ③ Infiltration of inflammatory cells: evaluated by the degree of leukocyte infiltration in alveolar and interstitial regions. ④ Thickening of the alveolar walls: based on interstitial edema and cellular proliferation. ⑤ Hyaline membrane formation: scored according to presence and extent. Each parameter is typically scored on a scale from 0-normal, 1-mild, 2-moderate, 3-severe, and 4-maximal damage, with the total lung injury score obtained by summing the individual parameter scores, providing an overall assessment of injury severity [38].

### 2.6. The Lung Wet/Dry (W/D) Weight Ratio

The lungs were carefully excised and blotted gently with filter paper to remove excess surface blood and moisture. The wet weight was recorded using an analytical balance. The lungs were then dried in an oven at 60 °C for 48 h until a constant weight was achieved. The dry weight was measured with the same balance. The W/D ratio was calculated by dividing the wet weight by the dry weight.

### 2.7. Cell Viability and Injury Determination

Cell viability is typically assessed using assays that distinguish live cells from dead or compromised cells. Cell viability was assessed using Cell Counting Kit-8 (CCK-8, Dojindo, Tokyo, Japan) in accordance with the manufacturer’s instructions. Following the hypoxia reoxygenation culture (H/R), cells were incubated in 10 μL CCK-8 solution for another 3 h at 37 °C. The microplate spectrophotometer was used to measure the optical density (OD) values at 450 nm. Elevated LDH levels in the culture medium indicate compromised cell membranes, reflecting cell injury. LDH, TNF-α, and IL-1β were assessed by a commercial assay kit.

### 2.8. Fe^2+^, MDA, 4-HNE, and GSH Measurements

The levels of Fe^2+^, MDA, 4-HNE, and GSH in lung tissues or cells were measured using a commercial iron assay kit, MDA test kit, 4-HNE ELISA kit, and GSH assay kit that were purchased from Nanjing Jiancheng Bioengineering Institute (Nanjing, China) in accordance with the manufacturer’s instructions, respectively.

### 2.9. Western Blotting

Western blotting is a widely used technique to detect and quantify specific proteins within a sample, which is used to assess the expression levels of proteins involved in ferroptosis, signaling pathways, and other relevant markers. The expression levels of protein TFRC, ACSL4, and the PKA-Hippo-YAP signaling pathway were assessed by Western blotting analysis. Briefly, lung tissues or cells were lysed in RIPA buffer containing protease inhibitors after homogenization. The total quantity of protein was calculated using a BCA protein assay kit. Proteins were separated by SDS-PAGE and then transferred to PVDF membranes. After blocking with 5% non-fat milk, the membranes were incubated with primary antibodies against TFRC, ACSL4, p-PKA/PKA, p-LATS1/2/LATS1/2, p-YAP/YAP, and β-actin or GAPDH for an entire night. After that, the membranes were incubated with secondary antibodies and imaged using an ECL detection system.

### 2.10. Lipid ROS Detection

Lipid reactive oxygen species (ROS) are highly reactive molecules generated during lipid peroxidation, a hallmark of ferroptosis. DCFH-DA is a cell-permeable, non-fluorescent probe commonly used to measure overall intracellular ROS levels. Add 1 mL of the 10 µM DCFH-DA solution to each well and incubate the plate at 37 °C for 30 min in the dark. After incubation, remove the DCFH-DA solution and wash cells twice with PBS to remove excess dye. Collect cells by centrifugation for 5 min and resuspend in PBS. Transfer the cell suspension to a flow cytometry tube and measure fluorescence using a flow cytometer (Cytofex, Beckman Coulter, Brea, California, USA) with excitation at 488 nm and emission at 530 nm. Normalize data to cell number or protein content to account for variations in cell density. In the histogram, each cell is represented along the x-axis by its fluorescence intensity. The Y-axis represents the number of events (cell count) at each fluorescence intensity. So it’s the frequency of cells.

### 2.11. Transmission Electron Microscopy

Lung tissue samples or cells were fixed in a solution containing 2.5% glutaraldehyde and 2% paraformaldehyde in 0.1 M cacodylate buffer. The tissues were incubated with uranyl acetate and lead citrate and then cut into 50–70 nm ultrathin sections using a Leica Ultracut microtome. After then, the ultrastructure morphology of mitochondria was examined using a transmission electron microscope (Japan Transmission Electron Microscope JEOL1010, Akishima, Tokyo, Japan).

### 2.12. Statistical Analysis

All data were recorded as mean ± standard deviation (SD), and SPSS 22.0 (Chicago, IL, USA) software was applied for statistical analysis. Statistical comparisons among groups were analyzed using one-way analysis of variance (ANOVA) followed by Bonferroni’s multiple comparison test. Statistical significance was set at *p* < 0.05.

## 3. Results

### 3.1. MaR1 Alleviates LIRI and Ferroptosis After Lung Transplantation

To explore the role of MaR1 on ferroptosis in LIRI, mice were exposed to MaR1 by intravenous injection 30 min before reperfusion. As illustrated in Figure 1A,B, compared with the control group, ischemia reperfusion dramatically led to pathological damage in the lung, as indicated by increased lung injury scores, which were determined by alveolar edema, hemorrhage, and inflammatory cell infiltration. However, MaR1 administration effectively alleviated these changes, as shown by decreased lung injury scores and wet/dry ratio. Furthermore, the relative content of proinflammatory cytokines, including TNF-α and IL-1β, was markedly increased in the LIRI group relative to the control group, whereas these levels were significantly decreased by MaR1 treatment (Figure 1B).

Compared with the control group, Fe^2+^, lipid peroxidation MDA, and 4-HNE were increased in the LIRI group, accompanied by a remarkable decrease in GSH level. MaR1 treatment decreased ischemia reperfusion stimulated Fe^2+^, MDA, and 4-HNE and restored the GSH levels (Figure 1C). TFRC and ACSL4 are prominent ferroptosis biomarkers. As shown in Figure 1D, the upregulation of TFRC and ACSL4 was consistently induced in the LIRI group as compared with the control group. These changes were significantly reversed by the MaR1 administration. Mitochondrial morphology under a transmission electron microscope is another important indicator of ferroptosis. In Figure 1E, ischemia reperfusion-induced mitochondrial shrinkage, mitochondrial cristae disappearance, and a decrease in the mitochondrial bilayer membrane density, which was repaired by MaR1 treatment. Overall, these results suggest that ferroptosis is implicated in the process of LIRI development, and MaR1 can alleviate lung injury after lung transplantation.

### 3.2. MaR1 Ameliorates LIRI by Inhibiting Ferroptosis

In order to verify the effect of MaR1 on ferroptosis in LIRI, ferroptosis agonist sorafenib was applied in vivo. As shown in Figure 2A,B, sorafenib significantly aggravated lung injury score, wet/dry ratio, and proinflammatory factors (TNF-α and IL-1β) release compared with the LIRI group. However, this sorafenib-induced lung injury was partly repressed by MaR1 administration. Compared with the LIRI group, Fe^2+^, MDA, and 4-HNE were increased in the LIRI + Sora group, accompanied by a remarkable decrease in GSH level. MaR1 treatment inhibited sorafenib-induced Fe^2^, MDA, and 4-HNE and restored the GSH levels (Figure 2C). Ischemic reperfusion-induced mitochondrial shrinkage, mitochondrial cristae disappearance, and a decrease in the mitochondrial bilayer membrane density were worsened by sorafenib, which was partly rescued by MaR1 treatment (Figure 2D).

### 3.3. MaR1 Dampens Ferroptosis via Activating the PKA/LATS1/2/YAP Pathway in LIRI

We next explored the potential mechanism involved in the influence of MaR1 on ferroptosis in vivo. Previous studies have shown that TFRC and ACSL4 expression in ferroptosis are regulated by the Hippo/YAP signaling pathway, which can also be activated by PKA phosphorylation. The PKA phosphorylation pathway is a downstream intracellular signaling cascade of LGR6 (a G-protein-coupled receptor), which is the binding receptor of MaR1. Therefore, we explored whether MaR1 activates the Hippo/YAP pathway via PKA phosphorylation to repress ferroptosis and manipulated it using specific inhibitors. As shown in Figure 3A–C, the protein expression levels of p-PKA, p-LATS1/2/ and p-YAP in the LIRI group were obviously inhibited compared with those in the control group. Importantly, MaR1 treatment significantly enhanced the p-PKA, p-LATS1/2, and p-YAP expression compared with the LIRI group.

Furthermore, it was observed that the lung tissues were more severely damaged and the inflammatory factors (TNF-α and IL-1β) were more intense in the LIRI+MaR1+KT5720 group and LIRI+MaR1+verteporfin group than the LIRI+MaR1 group (Figure 3D,E). Consistently, KT5720 or verteporfin application abolished the MaR1-mediated downregulation of TFRC and ACSL4 (Figure 3F,G). Compared with the LIRI+MaR1 group, the KT5720 or verteporfin application significantly increased Fe^2+^, MDA, 4-HNE, and decreased GSH levels (Figure 3H). Typical ferroptosis mitochondria with reduced cristae, shrunken volume, and increased membrane density were more abundant in the LIRI+MaR1+KT5720 group and LIRI+MaR1+verteporfin group compared with the LIRI+MaR1 group (Figure 3I). These results suggest that MaR1 dampens ferroptosis by activating the PKA-Hippo-YAP signaling pathway in LIRI.

### 3.4. MaR1 Inhibits Hypoxia/Reoxygenation-Induced Ferroptosis In Vitro

Pulmonary endothelial cells, a component of the alveolar air-blood barrier, play a critical role in the pathogenesis process of LIRI. Therefore, we further determined the effect of MaR1 on ferroptosis in Human Pulmonary Microvascular Endothelial Cell Lines (HPMEC) after H/R culture. The cell viability was decreased, and cell injury was exacerbated in the H/R group compared to the control group. Notably, the cell viability was evidently enhanced and cell injury was attenuated by MaR1 treatment in HPMEC cells after H/R culture (Figure 4A,B).

The influence of MaR1 on ferroptosis was further verified by ferroptosis hallmarks. As illustrated in Figure 4C–F, H/R accelerated ferroptosis in HPMEC cells as indicated by increased Fe^2+^, MDA, and ROS and decreased GSH levels compared to the control group. Simultaneously, MaR1 treatment decreased the Fe^2+^, MDA, and ROS levels while raising the GSH content as compared with the H/R group. Meanwhile, TEM analysis showed that H/R resulted in mitochondrial shrinkage, cristae disappearance, and decreased bilayer membrane density; however, these features were not observed in H/R+MaR1 treatment (Figure 4G). In Figure 4H,I, H/R challenge resulted in an obvious increase in TFRC and ACSL4 expression, which were reversed by MaR1 intervention.

Next, ferroptosis agonist sorafenib was applied to verify the effect of MaR1 on ferroptosis in vitro. As shown in Figure 5A,B, sorafenib significantly aggravated cell death and injury compared to the H/R group. However, these sorafenib-induced cell death and injury were reversed by MaR1 administration. In addition, sorafenib obviously enhanced the Fe^2+^, MDA, and ROS levels and reduced the GSH content compared to the H/R group, while this effect was restored by MaR1 treatment (Figure 5C,E). H/R-induced mitochondrial shrinkage, mitochondrial cristae disappearance, and a decrease in the mitochondrial bilayer membrane density were aggravated by sorafenib, which was partly rescued by MaR1 treatment (Figure 5D). Taken together, these data demonstrated that MaR1 protects against H/R-induced ferroptosis in HPMEC cells.

### 3.5. MaR1 Suppresses Ferroptosis by PKA/LATS1/2/YAP Activation In Vitro

We further investigated whether MaR1 phosphorylates PKA to activate the Hippo signaling pathway in H/R-induced HPMEC cells. It was found that MaR1 treatment increased the protein expression of p-PKA, p-LATS1/2, and p-YAP in the H/R+MaR1 group than the H/R group. Moreover, KT5720 restrained the upregulation effect of MaR1 on p-PKA, p-LATS1/2, and p-YAP levels, whereas verteporfin only abolished MaR1-induced YAP phosphorylation (Figure 6A,B).

The suppressive effects of MaR1 on LDH, Fe^2+^, MDA, ROS level, TFRC, and ACSL4 expression, as well as the promoting effect of MaR1 on cell viability and GSH content in H/R-induced HPMEC cells, were reversed by KT5720 or verteporfin application (Figure 6C–G). Typical ferroptosis mitochondria with reduced cristae, shrunken volume, and increased membrane density were more abundant in the H/R+MaR1+KT5720 group and H/R+MaR1+verteporfin group as compared with the H/R+MaR1 group (Figure 6B). These results demonstrated that the PKA-Hippo-YAP signaling pathway is responsible for the repression effect of MaR1 on H/R-induced ferroptosis in HPMEC cells (Figure 6H).

## 4. Discussion

The imbalance of iron homeostasis, lipid peroxidation, and redox synthesis is closely linked with the onset of ferroptosis. Abnormal accumulation of Fe^2+^ appeared to increase with time after lung transplantation, which causes excessive ROS generation through the Fenton reaction and amplifies the activity of multiple enzymes critical for redox and lipid metabolism [39,40,41]. Lu et al. ever reported that ferroptosis is triggered in LIRI and that inhibiting ferroptosis with ferrostatin-1 can prohibit lipid peroxidation, inflammation, and apoptosis in the lungs [42]. Li and Zhao et al. demonstrated that ferroptosis is overactivated in LIRI and that blocking ferroptosis with liproxstatin-1 can decrease iron accumulation, lipid peroxidation, and inflammation in lung tissues [43,44]. Our studies, both in vivo and in vitro, revealed that Fe^2+^ concentration, along with TFRC and ACSL4 levels, was elevated following LIRI or H/R but was decreased after MaR1 treatment. Lipid peroxidation, a free radical reaction, leads to the synthesis of various end products of lipid metabolism that result in cellular damage. GSH is a crucial intracellular antioxidant that hinders ferroptosis by neutralizing lipid peroxide derivatives and stabilizing the cell membrane from oxidative injury [45,46,47]. In the current study, we observed that MaR1 decreased ROS, MDA, and 4HNE levels, thereby ameliorating lung tissue and cell damage caused by ischemia reperfusion. Additionally, GSH levels were also markedly declined after LIRI or H/R but were restored by MaR1. These findings demonstrated that MaR1 could mitigate LIRI by inhibiting ferroptosis.

MaR1, a specialized pro-resolving mediator derived from docosahexaenoic acid, has been proven to exert inflammation-resolving and tissue-protective properties in lung, liver, hippocampal neuroinflammation and cognitive function, sepsis-induced microglial neuritis, and kidney injury by modulating inflammation, oxidative stress, and apoptosis. MaR1’s anti-ferroptotic actions—observed here in LIRI and replicated in neuroinflammation and sepsis-induced microglial neuritis—suggest a conserved mechanism centered on redox balance and lipid metabolism. This consistency reinforces the plausibility of targeting the MaR1-ferroptosis axis as a universal strategy for other diseases [48,49,50,51]. Our previous study reported that MaR1 reduced lung injury scores, wet/dry weight ratios, myeloperoxidase activity, TNF-α levels, and pulmonary permeability indices in an orthotopic ischemia reperfusion injury model [52]. Moreover, we found that MaR1 ameliorated LIRI by suppressing oxidative stress via activation of the Nrf2-mediated HO-1 signaling pathway, leading us to investigate the potential role of MaR1 in effectively inhibiting ferroptosis. Another study also reported that MaR1 ameliorated liver injury by decreasing lipid peroxidation, iron accumulation, and ROS production in an acute liver injury model [53]. Additionally, Zhang et al. found that MaR1 prevented bone loss, enhanced bone microstructure, and increased bone mineral density in high-glucose-induced osteoporosis by inhibiting ferroptosis in osteoblasts and osteocytes [54]. Li et al. ever demonstrated that MaR1 protects adult retinal pigment epithelial cells from high glucose-induced ferroptosis by triggering the Nrf2/HO-1/GPX4 pathway, suggesting that MaR1 can prevent diabetic retinopathy development [55]. Nevertheless, all of them primarily concentrated on Nrf2 and its downstream targets, HO-1 and GPX-4, which are pivotal regulators of ferroptosis via the cellular antioxidant defense mechanism. While MaR1 demonstrates protective effects consistent with ferroptosis inhibition, it is important to note that it also possesses anti-inflammatory and antioxidant properties, which may contribute to its overall efficacy. We acknowledge that additional mechanisms, including anti-inflammatory and antioxidant pathways, can initiate ferroptosis and ischemia-reperfusion injury, and they can interact with each other and further exacerbate ischemia-reperfusion injury, which is difficult to completely distinguish. Further studies with drugs that selectively act on ferroptosis are warranted to delineate these contributions.

PKA, a cyclic adenosine monophosphate (cAMP)-dependent serine/threonine kinase, is activated when upstream of G-protein-coupled receptors (GPCRs) coupled to Gαs subunits. PKA is responsible for phosphorylating a variety of substrates, thereby modulating gene expression, metabolic processes, and cell cycle progression [56]. Leucine-rich repeat containing-GPCR 6 (LGR6), a member of the GPCR family on the cell membrane surface, is identified as the receptor for MaR1 [57]. Additionally, GPCR receptors can also regulate the Hippo signaling pathway through diverse signal transduction mechanisms, including PKC, PKA, and Rho GTPases [58]. The core components of this pathway contain mammalian STE20-like kinase 1/2 (MST1/2), large tumor suppressor 1/2 (LATS1/2/2), WW domain of the Sav family containing protein 1 (SAV1), MOB kinase activator 1 (MOB1), Yes-associated protein (YAP) or transcriptional coactivator with PDZ-binding motif (TAZ), and members of the TEA domain (TEAD) family. YAP and TAZ serve as the primary downstream effectors of this pathway, acting as transcriptional co-activators or co-repressors based on their interaction partners [59]. The modulation of YAP/TAZ activity appears to be a crucial factor in the cellular response to ischemia, affecting cell survival, proliferation, and apoptosis. Ischemic conditions have been proved to activate the Hippo pathway, leading to the phosphorylation and subsequent inactivation of YAP and TAZ [60]. For instance, Zhou et al. proved that the Hippo signaling pathway was turned on during hepatic ischemia-reperfusion injury to suppress its downstream effectors YAP/TAZ, which reduced cell death and inflammation and improved the prognosis of liver surgery [61]. Matsuda et al. reported that oxidative stress-induced NF2 conferred cardio-protection through Mst1 activation and inhibition of YAP in myocardial ischemia reperfusion injury [62]. These studies highlight the dual role of the Hippo signaling pathway in ischemia reperfusion, which is involved in the modulation of oxidative stress, inflammation, and apoptosis.

Here, MaR1 was found to enhance the PKA activity and further induce the LATS1/2 and YAP phosphorylation, which in turn inhibits the nuclear localization and transcriptional activation of YAP, ultimately reducing the TFRC and ACSL4 expression. The use of a PKA inhibitor and a LATS1/2 antagonist, verteporfin, negated the inhibitory effect of MaR1 on TFRC and ACSL4 expression, suggesting that MaR1 represses ferroptosis through the PKA/LATS1/2/YAP signaling pathway, as demonstrated in both in vivo and in vitro studies. Notably, Martens et al. reported that sorafenib exhibited tissue-protective effects in an inflammatory disease model, highlighting its complex pharmacological profile. In our study, sorafenib’s ability to induce ferroptosis appears to contribute significantly to lung injury under hypoxia-reoxygenation conditions; however, these other effects cannot be entirely excluded. The observed protective effects of MaR1 in mitigating sorafenib-induced injury may involve modulation of multiple pathways, including ferroptosis and inflammation. Therefore, the activity of sorafenib in our model aligns with its multifaceted nature, and future investigations dissecting its specific contributions to ferroptosis versus other pathways will enhance our understanding of its role in lung injury [63]. Furthermore, while our pharmacological data suggest a PKA-YAP interaction, definitive causality requires genetic validation in vitro. Future studies will employ si-YAP RNA knockdown and adeno-associated virus (AAV9)-mediated PKA overexpression to confirm this axis in the pulmonary vascular endothelial cell line after H/R culture. Additionally, single-cell RNA sequencing could resolve cell-type-specific PKA-YAP crosstalk in pulmonary vascular endothelial cell-specific YAP-KO or PKA overexpression mouse models. We could also perform co-immunoprecipitation (Co-IP) to test PKA-Hippo-YAP Axis physical interactions. These revisions bridge the gap between pharmacological and genetic evidence while framing the current findings as part of an evolving mechanistic understanding. Overall, our findings unveil a novel mechanism by which MaR1 regulates ferroptosis and propose that MaR1 could be a promising candidate for the prevention and treatment of LIRI.

However, several limitations still need to be addressed in the next step. First, the lung transplantation model in mice was established in our study, which may not fully reflect the clinical situation of human lung transplantation, since there is always a species difference. While murine models identify fundamental pathways, their limitations necessitate validation in large animal models (e.g., pig, non-human primate) with lung physiology, immune responses, and coagulation systems closer to humans, and ideally, studies using human cells/tissues (e.g., EVLP, explants). Future research, especially in larger animals and human systems, must prioritize understanding and targeting the specific interplay between platelets, coagulation, complement, and inflammation in human LIRI. Mechanistically, the initiation of ferroptosis and inflammatory cascades occurs within hours of reperfusion, setting the stage for later tissue remodeling. Thus, targeting this window is critical for developing preconditioning or immediate post-treatment strategies. To assess the long-term implications of our findings, future studies will track survival, pulmonary function (e.g., airway resistance, compliance), and fibrosis (e.g., hydroxyproline content) over 7–28 days in a murine orthotopic lung transplant model. These endpoints align with clinical benchmarks for PGD resolution and chronic graft dysfunction. Second, the use of a single dose and time point of MaR1 was applied in this study, which should be optimized to achieve the ideal protective effect. A similar dosage was employed and reported significant benefits without adverse effects. We acknowledge that the use of a single, fixed dose and administration at a single time point represents limitations, and that the efficacy could be dose-dependent. Future studies should explore various doses and administration frequencies to optimize therapeutic outcomes. Targeting pathways before the moment of reperfusion offers the greatest potential for preventing the initiation and amplification of injury. This aligns perfectly with clinical opportunities during donor management, EVLP, SCS flush, and the implantation phase. Post-op strategies are vital for managing established PGD but are inherently less effective at blocking the initial wave of LIRI damage. They should be viewed as complementary in the future study. Third, the potential off-target effects of the pharmacological inhibitors were neglected; therefore, more precise and reliable methods could be used to manipulate the PKA/LATS1/2/YAP pathway. This study enriches a novel regulatory mechanism of ferroptosis in LIRI and exhibits that MaR1 may be a promising therapeutic approach. Considering that ferroptosis is involved in the pathological process of many diseases and that MaR1 effectively inhibits ferroptosis in our LIRI model, we may also anticipate its therapeutic potential in numerous ferroptosis-related diseases. Fourth, while our model employed rigorous controls for confounding factors (e.g., surgical trauma, temperature), the initial analysis did not include ischemia-specific molecular markers. Measurements of HIF-1α and LDH in the lung tissue can be used to confirm the ischemic component of ischemia reperfusion injury. Future studies could further delineate ischemia reperfusion injury from generalized stress by including additional markers (e.g., HIF-1α and LDH) or time-course analyses of hypoxia-sensitive pathways.

In this study, we utilized HPMECs to investigate endothelial responses to hypoxia/reoxygenation injury, given that pulmonary endothelial dysfunction is a critical early event in LIRI and plays a central role in modulating subsequent inflammatory and tissue damage responses. The choice of HPMECs was also driven by their availability and well-characterized nature for in vitro vascular studies. However, we acknowledge that LIRI involves complex interactions among multiple cell types, including alveolar epithelial cells, alveolar macrophages, and other immune cells, which contribute to the overall injury process. Future studies should incorporate co-culture systems or primary cell cultures of alveolar epithelial cells and immune cells to better model the cellular interplay and to fully understand MaR1’s effects within the lung microenvironment.

## 5. Conclusions

In the present study, we have shown that MaR1 can effectively ameliorate LIRI mechanically by impeding ferroptosis. The application of antagonists KT5720 and verteporfin hampered the protective effect of MaR1 on LIRI and reversed the MaR1-induced downregulation of TFRC and ACSL4. Moreover, our in vivo studies were consistent with the outcomes observed in the in vitro experiment, indicating that MaR1 mitigated LIRI by repressing ferroptosis, which is involved in the PKA/LATS1/2/YAP signaling pathway (depicted in Figure 7).

## Figures and Tables

**Figure 1 biomedicines-13-01594-f001:**
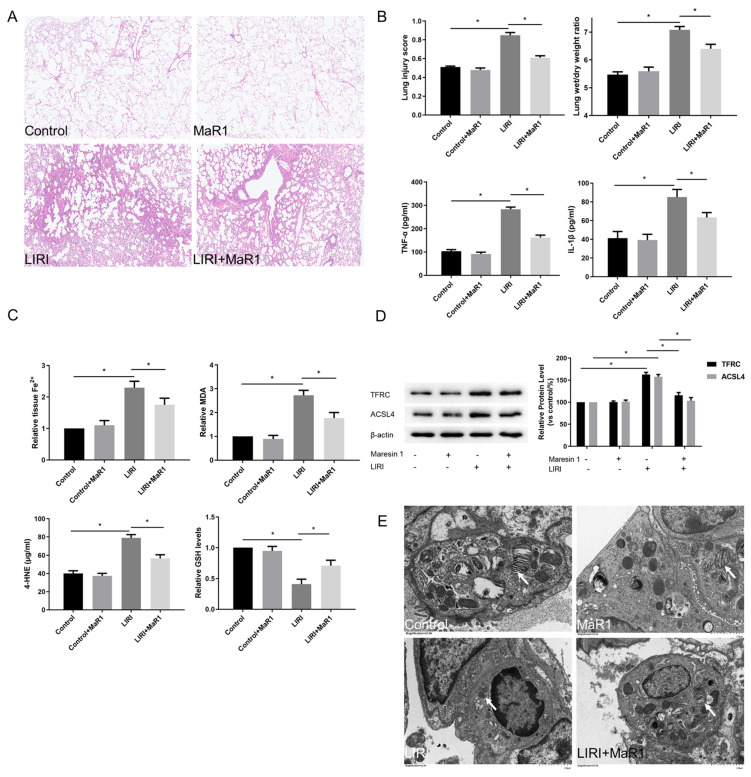
MaR1 alleviates LIRI and ferroptosis after lung transplantation. (**A**) Representative H&E staining of lung tissues (original magnification, ×200). (**B**) Acute lung injury score and wet/dry ratio of each group, and the relative contents of the proinflammatory cytokines TNF-α and IL-1β in lung tissue. (**C**) Relative values of Fe^2+^, MDA, 4-HNE, and GSH concentrations. (**D**) Representative western blotting and quantification analysis of TFRC and ACSL4 expression. (**E**) Representative TEM images of each group. The white arrow indicates ferroptotic mitochondria, magnification ×8000. Data are presented as the mean ± SD, *n* = 6. * *p* < 0.05.

**Figure 2 biomedicines-13-01594-f002:**
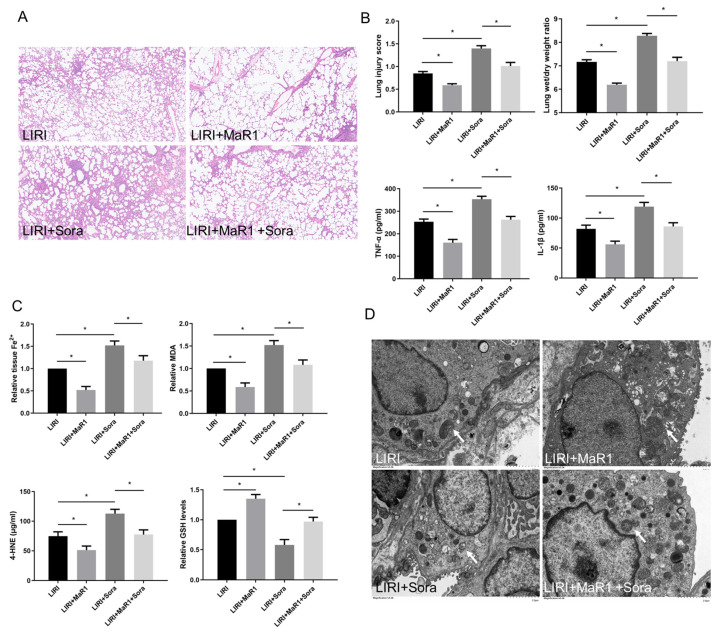
MaR1 ameliorates LIRI by inhibiting ferroptosis. (**A**) Representative H&E staining of lung tissues (original magnification, ×200). (**B**) Acute lung injury score and wet/dry ratio of each group, and the relative levels of the proinflammatory cytokines TNF-α and IL-1β in lung tissue. (**C**) Relative values of Fe^2+^, MDA, 4-HNE, and GSH concentrations. (**D**) Representative TEM images of each group. The white arrow indicates ferroptotic mitochondria, magnification ×8000. Data are presented as the mean ± SD, *n* = 6. * *p* < 0.05.

**Figure 3 biomedicines-13-01594-f003:**
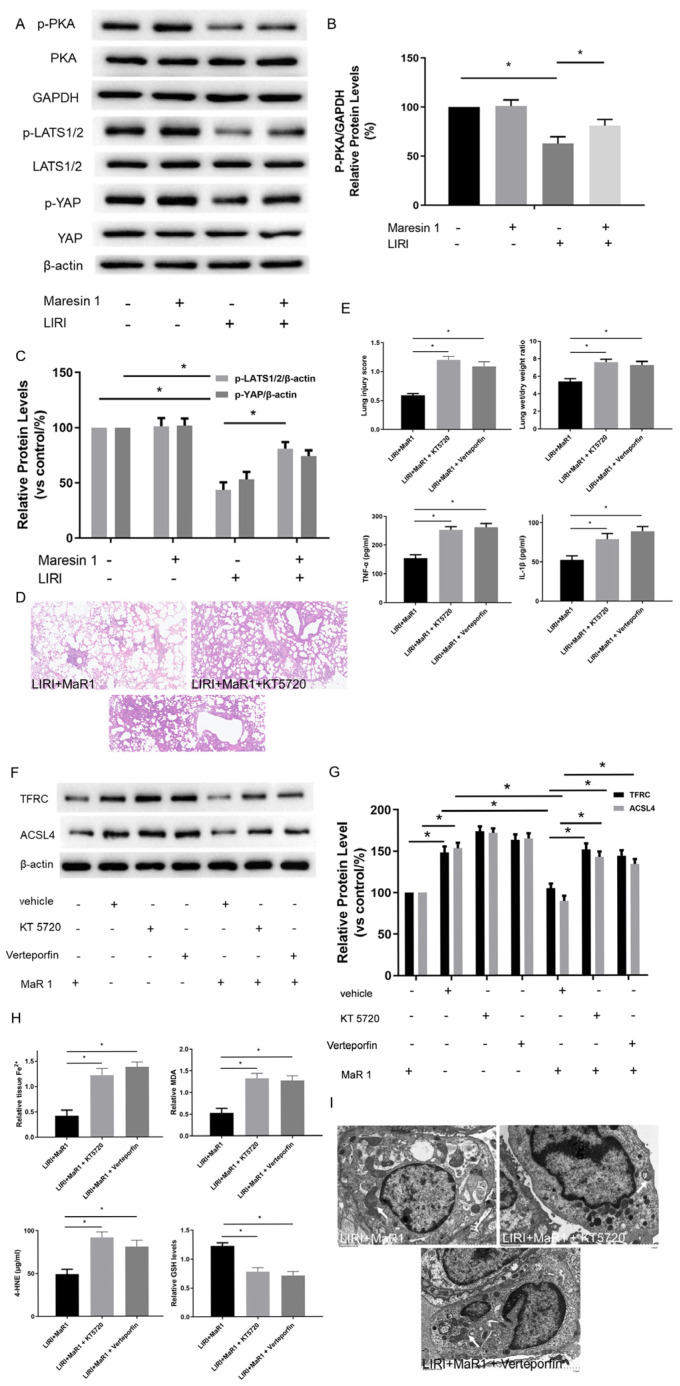
MaR1 dampens ferroptosis via activating the PKA/LATS1/2/YAP pathway in LIRI. (**A**) The protein expression levels of the PKA/LATS1/2/YAP pathway were determined by western blotting. (**B**) Quantification analysis of the protein expression levels of p-PKA. (**C**) Quantification analysis of the expression levels of p-LATS1/2 and p-YAP protein. (**D**) Representative H&E staining of lung tissues (original magnification, ×200). (**E**) Acute lung injury score and wet/dry ratio of each group, and the relative contents of the proinflammatory cytokines TNF-α and IL-1β in lung tissue. (**F**) Representative western blotting results of TFRC and ACSL4 expression. (**G**) Quantification analysis of TFRC and ACSL4 expression levels. (**H**) Relative values of Fe^2+^, MDA, 4-HNE, and GSH concentrations. (**I**) Representative TEM images of each group. The white arrow indicates ferroptotic mitochondria, magnification ×8000. Data are presented as the mean ± SD, *n* = 6. * *p* < 0.05.

**Figure 4 biomedicines-13-01594-f004:**
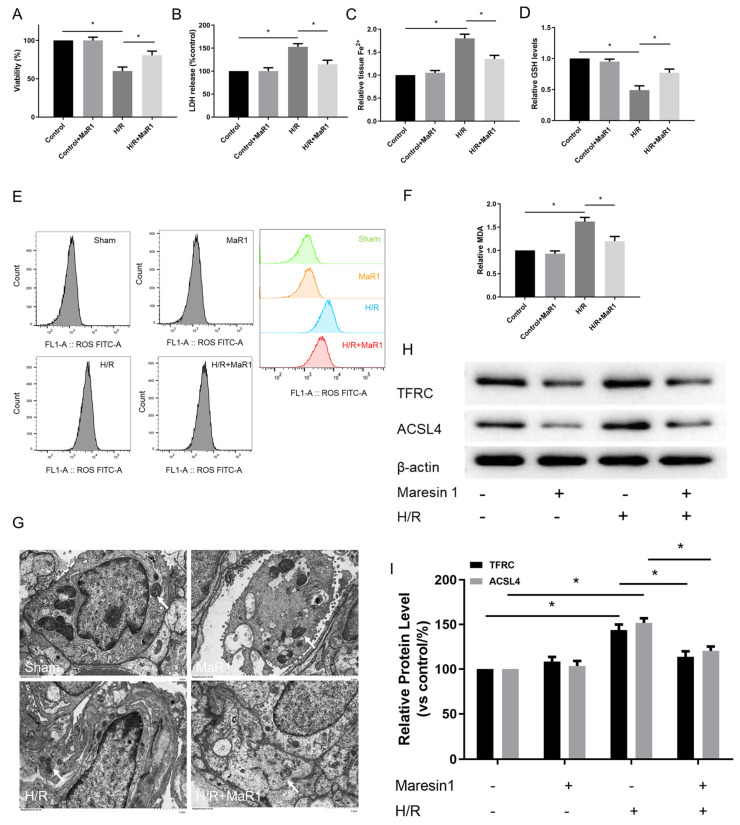
MaR1 inhibits hypoxia/reoxygenation-induced ferroptosis in vitro. (**A**) Fold change in cell viability of the HPMEC cells. (**B**) Measurement of cell injury marker LDH in the HPMEC cells. (**C**) Relative values of Fe^2+^ concentrations. (**D**) Relative values of GSH concentrations. (**E**) The levels of ROS were determined with the DCFH-DA fluorescent probe and quantified by flow cytometry. (**F**) Relative values of MDA concentrations. (**G**) Representative TEM images of each group. The white arrow indicates ferroptotic mitochondria, magnification ×8000. (**H**) Representative western blotting results of the TFRC and ACSL4 expression. (**I**) Quantification analysis of the TFRC and ACSL4 expression levels. Data are presented as the mean ± SD, *n* = 6. * *p* < 0.05.

**Figure 5 biomedicines-13-01594-f005:**
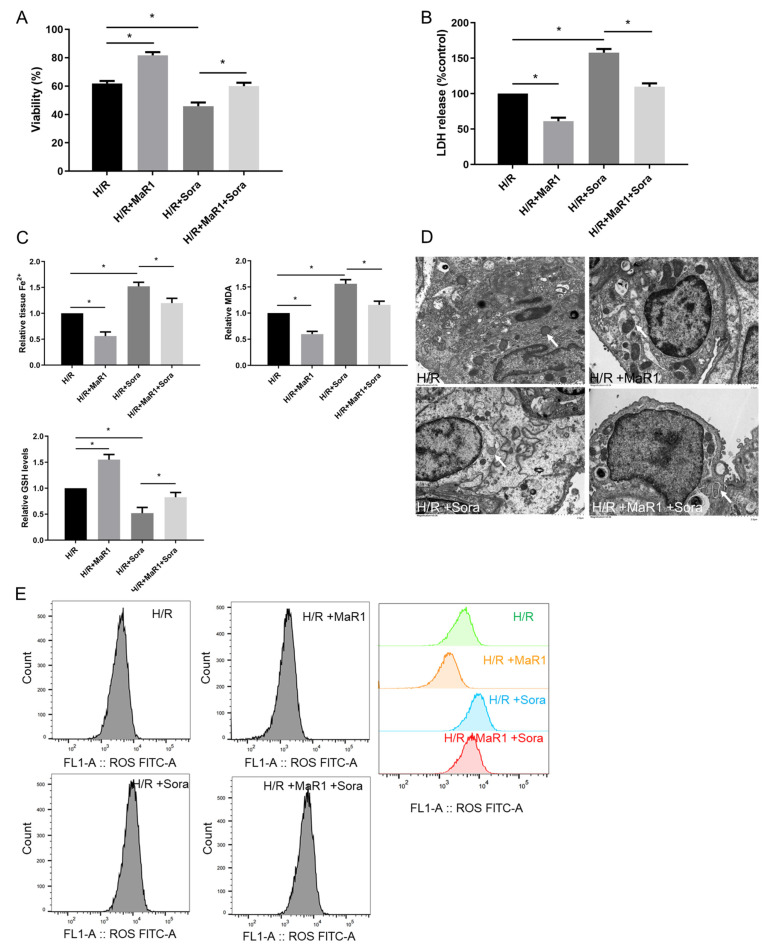
Ferroptosis agonist sorafenib was applied to verify the effect of MaR1 on ferroptosis in vitro. (**A**) Fold change in cell viability of the HPMEC cells. (**B**) Measurement of cell injury marker LDH in the HPMEC cells. (**C**) Relative values of Fe^2+^, MDA, and GSH concentrations. (**D**) Representative TEM images of each group. The white arrow indicates ferroptotic mitochondria, magnification ×8000. (**E**) The levels of ROS were determined with the DCFH-DA fluorescent probe and quantified by flow cytometry. Data are presented as the mean ± SD, *n* = 6. * *p* < 0.05.

**Figure 6 biomedicines-13-01594-f006:**
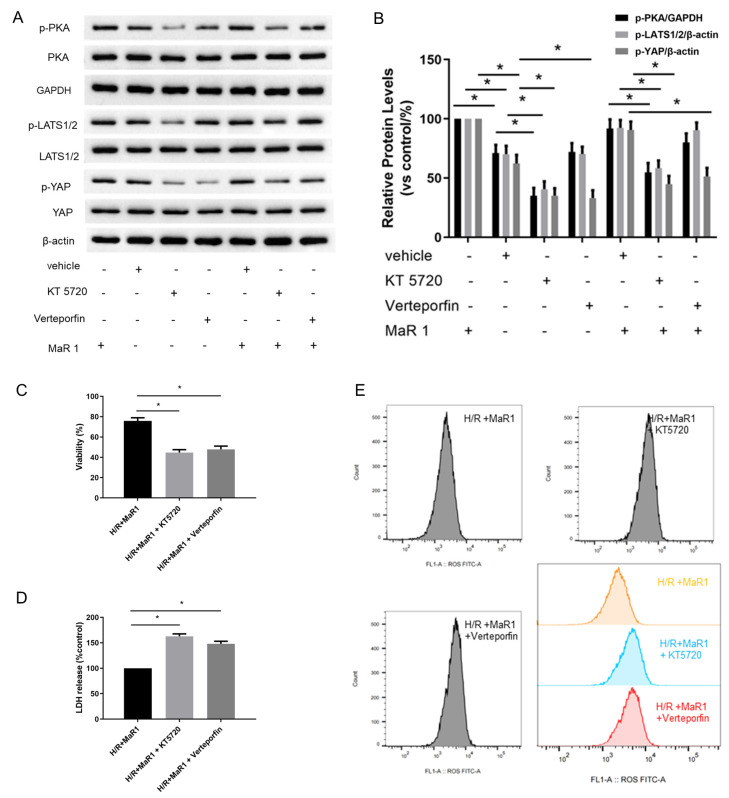
MaR1 suppresses ferroptosis by PKA/LATS1/2/YAP activation in vitro. (**A**) The protein expression levels of the PKA/LATS1/2/YAP pathway were determined by western blotting. (**B**) Quantification analysis of the protein expression levels of p-PKA, p-LATS1/2, and p-YAP protein. (**C**) Fold change in cell viability of the HPMEC cells. (**D**) Measurement of cell injury marker LDH in the HPMEC cells. (**E**) The levels of ROS were determined with the DCFH-DA fluorescent probe and quantified by flow cytometry. (**F**) Representative western blotting results of TFRC and ACSL4 expression. (**G**) Quantification analysis of the TFRC and ACSL4 expression levels. (**H**) Representative TEM images of each group. The white arrow indicates ferroptotic mitochondria, magnification ×8000. Data are presented as the mean ± SD, *n* = 6. * *p* < 0.05.

**Figure 7 biomedicines-13-01594-f007:**
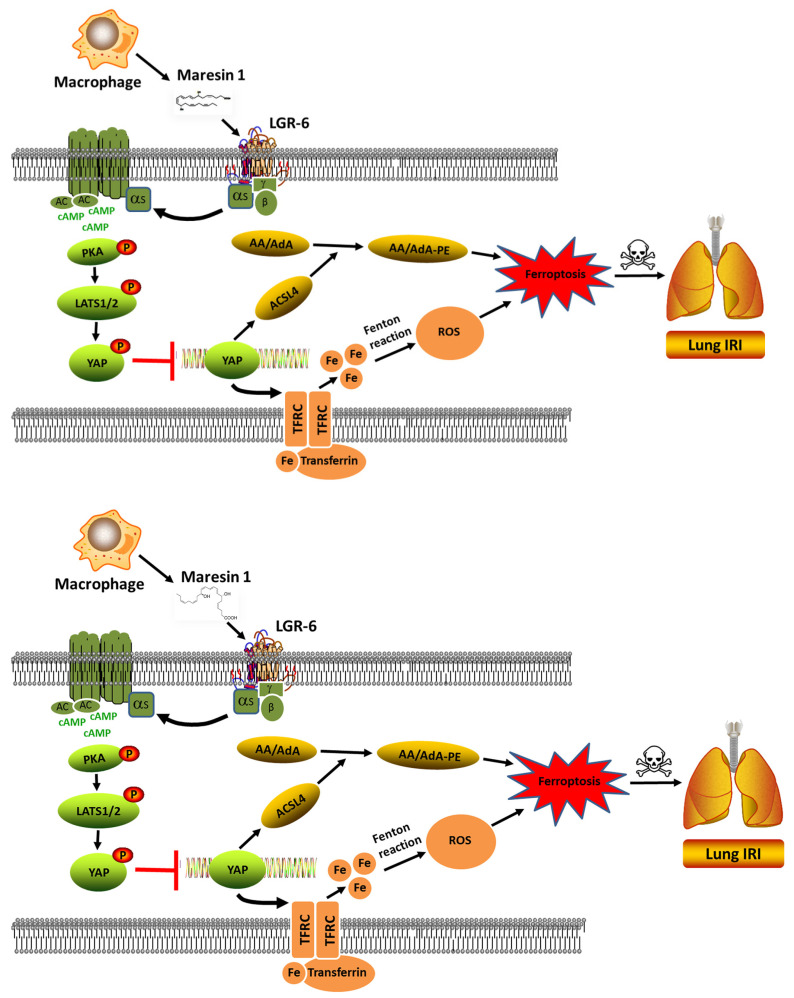
The schematic figure for MaR1 mitigated LIRI by repressing ferroptosis through the PKA/LATS1/2/YAP signaling pathway.

## Data Availability

The data and materials that support the study are available from the corresponding author on reasonable request.

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
