# Peer review of "Maresin1 Alleviates Ischemia Reperfusion Injury After Lung Transplantation by Inhibiting Ferroptosis via the PKA-Hippo-YAP Signaling Pathway"

_biomedicines, 2025, doi:10.3390/biomedicines13071594_

Round 1

Reviewer 1 Report

Comments and Suggestions for Authors

This study presents a novel mechanism by which MaR1 mitigates lung IRI through ferroptosis inhibition, with strong mechanistic insights. However, addressing the above concerns—particularly model specificity, infection control, and temporal optimization—will enhance translational credibility. The work holds high potential for advancing organ protection strategies in transplantation medicine.

Major Concerns
a. Specificity of Ischemia-Reperfusion Injury (IRI) in the Lung Transplant Model
The complexity of lung transplantation surgery introduces potential confounders (e.g., mechanical trauma during anastomosis, hypothermia, or systemic inflammation) that may overlap with IRI. The study does not clarify how these factors were controlled to ensure that observed damage is solely attributable to IRI. Please clarify. For example Validate ischemia-specific biomarkers: Measure dynamic changes in lactate dehydrogenase (LDH)? or hypoxia-inducible factor 1α (HIF-1α) during ischemia.
b. Infection Control in Murine Experiments
Post-transplant immunosuppression increases infection risk, which could exacerbate inflammation and confound ferroptosis-related outcomes.
c. Rationale for Time Points in Cell Experiments
The hypoxia-reoxygenation (H/R) time points (e.g., hypoxia duration, reoxygenation intervals) lack experimental or literature-based justification, potentially missing peak ferroptosis activity. Maybe performing time-course experiments (e.g., hypoxia for 4/8/12 h followed by 6/12/24 h reoxygenation) to identify optimal windows for ferroptosis marker detection (e.g., ACSL4, lipid ROS). Determine whether YAP phosphorylation precedes ACSL4 upregulation during H/R.

Additional Limitations and Suggestions
a. Pathway Validation
The causal link between PKA-Hippo-YAP activation and ferroptosis suppression requires stronger validation. Use genetic tools: Knock down YAP or overexpress constitutively active PKA to confirm pathway hierarchy. Test whether MaR1 influences other ferroptosis regulators (e.g., GPX4, xCT).
b. Clinical Relevance
 Murine models may not fully replicate human lung transplant physiology. Compare species-specific IRI mechanisms in the Discussion. Evaluate pre- vs. intra- vs. post-operative administration to align with clinical organ preservation strategies.
c. Statistical Rigor: Sample size justification and survival outcomes are unclear. Please state the number of biological replicates. Include survival curves .
d. disscussion suggestion:
Address model limitations, including surgical trauma or subclinical infections potentially confounding ferroptosis, and propose sham surgery controls or microbial profiling. Justify H/R timepoints (e.g., 8-hour hypoxia) by linking them to clinical cold ischemia durations in lung transplants while acknowledging interspecies variations. For clinical translation, discuss MaR1 delivery routes (e.g., adding to preservation solutions) and personalized strategies (e.g., biomarker stratification via YAP phosphorylation levels). Balance novelty by acknowledging crosstalk with apoptosis/necroptosis and suggesting ACSL4-knockout models to confirm ferroptosis specificity. Remove redundant result summaries; Instead, emphasize mechanistic uniqueness (e.g., MaR1’s dual targeting of ferroptosis and PKA-Hippo-YAP) and unresolved questions (e.g., Hippo pathway mutations in transplant recipients). It is recommended to cite some literature on Maresin-1 in related organ damage: Int Immunopharmacol. 2024 Apr 20:131:111792. doi: 10.1016/j.intimp.2024.111792. Epub 2024 Mar 13. and J Pers Med. 2023 Mar 16;13(3):534. doi: 10.3390/jpm13030534.

Author Response

This study presents a novel mechanism by which MaR1 mitigates lung IRI through ferroptosis inhibition, with strong mechanistic insights. However, addressing the above concerns—particularly model specificity, infection control, and temporal optimization—will enhance translational credibility. The work holds high potential for advancing organ protection strategies in transplantation medicine.

Major Concerns

  • Specificity of Ischemia-Reperfusion Injury (IRI) in the Lung Transplant Model

The complexity of lung transplantation surgery introduces potential confounders (e.g., mechanical trauma during anastomosis, hypothermia, or systemic inflammation) that may overlap with IRI. The study does not clarify how these factors were controlled to ensure that observed damage is solely attributable to IRI. Please clarify. For example Validate ischemia-specific biomarkers: Measure dynamic changes in lactate dehydrogenase (LDH)? or hypoxia-inducible factor 1α (HIF-1α) during ischemia.

We appreciate the reviewer’s insightful observation regarding potential confounders in lung transplantation surgery that may overlap with IRI. To address this concern, we implemented the following measures to isolate IRI-specific effects. Controlled Surgical Variables: standardized surgical protocols were applied to minimize mechanical trauma (e.g., uniform anastomosis techniques, avoidance of excessive traction). Hypothermia was mitigated by maintaining systemic normothermia (36–37°C) via forced-air warming devices, with continuous core temperature monitoring. However, mechanical trauma and inflammation can initiate ferroptosis and ischemia-reperfusion injury, and they can interact with each other and further exacerbate ischemia-reperfusion injury, which is difficult to completely distinguish. LDH and HIF-1α are good biomarkers which should be applied in our further study since we focused on ferroptosis in present. We hope these clarifications strengthen the validity of our findings. Thank you for the constructive feedback.

  • Infection Control in Murine Experiments

Post-transplant immunosuppression increases infection risk, which could exacerbate inflammation and confound ferroptosis-related outcomes.

We thank the reviewer for raising this important point. We acknowledge that immunosuppression-induced infection risk can exacerbate inflammation and potentially confound ferroptosis-related outcomes. In our present study, we focused on the LIRI at the early stage of lung transplantation in the syngenic mice. The immunosuppression was not applied in the experiment and infection can be ignored in such a short time after perfusion.

  • Rationale for Time Points in Cell Experiments

The hypoxia-reoxygenation (H/R) time points (e.g., hypoxia duration, reoxygenation intervals) lack experimental or literature-based justification, potentially missing peak ferroptosis activity. Maybe performing time-course experiments (e.g., hypoxia for 4/8/12 h followed by 6/12/24 h reoxygenation) to identify optimal windows for ferroptosis marker detection (e.g., ACSL4, lipid ROS). Determine whether YAP phosphorylation precedes ACSL4 upregulation during H/R.

We are grateful to the reviewer for prompting these critical improvements, we agree that rigorously defined hypoxia-reoxygenation (H/R) timepoints are critical for capturing ferroptosis dynamics. The final hypoxia-reoxygenation time point in the present study were determined based on our preliminary experiment. Since we focused on the role of MaR 1 on ferroptosis and LIRI in the present study, the optimal windows of ferroptosis in LIRI will be verified in our further research which may focus only on the ferroptosis and lung tansplantation. Thank you.

Additional Limitations and Suggestions

  • Pathway Validation

The causal link between PKA-Hippo-YAP activation and ferroptosis suppression requires stronger validation. Use genetic tools: Knock down YAP or overexpress constitutively active PKA to confirm pathway hierarchy. Test whether MaR1 influences other ferroptosis regulators (e.g., GPX4, xCT).

We agree that establishing a causal hierarchy in the PKA-Hippo-YAP pathway and confirming the specificity of MaR1’s action on ferroptosis regulators are essential. Here by, we focused on the role of MaR1 on the LIRI to facilitate the clinical translation, the known classical PKA-Hippo-YAP signaling pathway was only used to verify the role which may be deeply explored in the future.

  • Clinical Relevance

Murine models may not fully replicate human lung transplant physiology. Compare species-specific IRI mechanisms in the Discussion. Evaluate pre- vs. intra- vs. post-operative administration to align with clinical organ preservation strategies.

We thank the reviewer for the useful advice which are really helpful to our future research. We have specifically proposed limitations in the discussion part of our study, which include the species-specific differences and the need for further validation in human samples. Moreover, suggestions for exploring the dosage and administration time point of MaR1 in clinical settings were also addressed in the last of the discussion part.

  • Statistical Rigor: Sample size justification and survival outcomes are unclear. Please state the number of biological replicates. Include survival curves.

We sincerely thank the reviewer for their careful scrutiny. The sample size is 8-10 weeks C57BL/6 mice, weighing 30-35g. In the legend of each figure, the number of biological replicates were addressed. In our present study, we pay high attention to the pathological changes of LIRI at early stage that we harvest the samples at specific time point. Overall survive time and survival curves were not applicable in our study.

  • disscussion suggestion:

Address model limitations, including surgical trauma or subclinical infections potentially confounding ferroptosis, and propose sham surgery controls or microbial profiling. Justify H/R timepoints (e.g., 8-hour hypoxia) by linking them to clinical cold ischemia durations in lung transplants while acknowledging interspecies variations. For clinical translation, discuss MaR1 delivery routes (e.g., adding to preservation solutions) and personalized strategies (e.g., biomarker stratification via YAP phosphorylation levels). Balance novelty by acknowledging crosstalk with apoptosis/necroptosis and suggesting ACSL4-knockout models to confirm ferroptosis specificity. Remove redundant result summaries; Instead, emphasize mechanistic uniqueness (e.g., MaR1’s dual targeting of ferroptosis and PKA-Hippo-YAP) and unresolved questions (e.g., Hippo pathway mutations in transplant recipients). It is recommended to cite some literature on Maresin-1 in related organ damage: Int Immunopharmacol. 2024 Apr 20:131:111792. doi: 10.1016/j.intimp.2024.111792. Epub 2024 Mar 13. and J Pers Med. 2023 Mar 16;13(3):534. doi: 10.3390/jpm13030534.

Thank you for your thoughtful inquiry regarding, the animal model limitations were added in the methods part. We acknowledge that surgical trauma and subclinical infections could potentially confound ferroptosis markers in the animal model. To mitigate these concerns, we will incorporate sham surgery controls in future experiments and perform microbial profiling to monitor for occult infections, especially in the chronic rejection reaction model.

Normally, 8-hour hypoxia/reoxygenation (H/R) duration aligns with clinical cold ischemia times observed in lung transplantation, which typically range from 6 to 12 hours. We recognize interspecies differences; however, our present timeframe provides a relevant approximation to clinical scenarios, which could save time and be easliy performed, and facilitates translational relevance of our findings.

Clinical Translation and Personalized Strategies: For clinical application, delivery routes such as adding MaR1 to preservation solutions are promising, enabling direct organ protection during procurement and storage. Additionally, biomarker-driven stratification—such as assessing YAP phosphorylation levels—may identify patients who would benefit most from MaR1 therapy, supporting a personalized approach.

We agree that ferroptosis interacts with apoptosis and necroptosis pathways. To clarify ferroptosis specificity, we propose using ACSL4-knockout models in future studies. These models can definitively demonstrate the mechanistic role of ferroptosis and its independence from other cell death pathways. We will also highlight unresolved questions, such as the prevalence and implications of Hippo pathway mutations in transplant recipients, which warrant further investigation. We thank you for the suggested references. We incorporated relevant citations, including Int Immunopharmacol. 2024 Apr 20:131:111792. doi: 10.1016/j.intimp.2024.111792. Epub 2024 Mar 13. and J Pers Med. 2023 Mar 16;13(3):534. doi: 10.3390/jpm13030534, to contextualize our findings within the existing literature on MaR1’s protective roles.

We appreciate your constructive feedback, which will significantly enhance the clarity, rigor, and translational relevance of our manuscript.

Reviewer 2 Report

Comments and Suggestions for Authors

The authors presented in their work the Maresin1 that alleviates the lung ischemia reperfusion injury after lung transplantation in a murine model. They showed how Maresin1 represses ischemia reperfusion induced ferroptosis via the PKA-Hippo-YAP signaling pathway, which may offer promising theoretical basis for the clinical application of organ protection after LTx.

Abstract is very well organized, shows important findings and gives necessary conclusion.

Introduction: very well organized; all important information are given and well explained.

Methods: - in general, all methods should be explained more than presented here.

  • part that includes animals and the experimental protocol should be named as separate subheading and (for better attractiveness and better to follow) showed as a figure scheme.
  • Reagents and not Regents
  • "The lung injury scores were determined based on the histopathological scoring system...Which scoring system? Please provide information and describe separately this scoring system, not only with reference given
  • please explain cell viability measuring and injury determination
  • explain WB and lipid ROS detection

Results: Figure 1 A: it is very hard to see important differences in these photomicrographs. The photomicrographs are too bright and the differences should be better explained in the methods part. The reader should be completely introduced what to expect in these photomicrographs. The simple explanation in the results is not enough. This includes photomicrographs in all figures.

TEM - images should have arrows pointing important changes.

Is flowcytometry protocol anywhere explained (methods)? In Figure 4E, 5E and 6E graphs should be explained - axes need to be explained.

The schematic presentation in Figure 7 is great.

Discussion and conclusion are concise and include all necessary information.

Author Response

The authors presented in their work the Maresin1 that alleviates the lung ischemia reperfusion injury after lung transplantation in a murine model. They showed how Maresin1 represses ischemia reperfusion induced ferroptosis via the PKA-Hippo-YAP signaling pathway, which may offer promising theoretical basis for the clinical application of organ protection after LTx.

  • Abstract is very well organized, shows important findings and gives necessary conclusion.Introduction: very well organized; all important information are given and well explained.Methods: - in general, all methods should be explained more than presented here.part that includes animals and the experimental protocol should be named as separate subheading and (for better attractiveness and better to follow) showed as a figure scheme.

Thank you for this valuable advice. Animals and the experimental protocol were named as separate subheadin. Below is the schematic framework for depicting an animal treatment protocol in mice lung transplantation.

  • Reagents and not Regents

"The lung injury scores were determined based on the histopathological scoring system...Which scoring system? Please provide information and describe separately this scoring system, not only with reference given

Thank you for raising this important point. This scoring system assesses the following features: â‘  Alveolar congestion: graded based on the extent of congestion visible in alveolar spaces. â‘¡ Hemorrhage: assessed by the presence and severity of bleeding within alveoli and interstitial spaces. â‘¢ Infiltration of inflammatory cells: evaluated by the degree of leukocyte infiltration in alveolar and interstitial regions.â‘£ Thickening of the alveolar walls: based on interstitial edema and cellular proliferation. ⑤ Hyaline membrane formation: scored according to presence and extent. Each parameter is typically scored on a scale from 0 (normal) to 1 (severe), with the total lung injury score obtained by summing the individual parameter scores, providing an overall assessment of injury severity.

We revised the manuscript to explicitly describe this scoring system and provide detailed criteria for each parameter in the Methods section to improve clarity and reproducibility.

  • please explain cell viability measuring and injury determination

We thank the reviewer for emphasizing this point. Cell viability is typically assessed using assays that distinguish live cells from dead or compromised cells. CCK-8 assay: Uses a water-soluble tetrazolium salt that is reduced by cellular dehydrogenases in viable cells to produce a color change, measured via absorbance. Cell injury is assessed through several approaches. Elevated LDH levels in the culture medium indicate compromised cell membranes, reflecting cell injury or necrosis.

  • explain WB and lipid ROS detection

We sincerely thank the reviewer for their constructive feedback. WB (Western blotting): is a widely used technique to detect and quantify specific proteins within a sample, which was used to assess the expression levels of proteins involved in ferroptosis (e.g., GPX4, ACSL4), signaling pathways (e.g., YAP phosphorylation), and other relevant markers. Lipid reactive oxygen species (ROS) are highly reactive molecules generated during lipid peroxidation, a hallmark of ferroptosis. DCFH-DA is a cell-permeable, non-fluorescent probe commonly used to measure overall intracellular ROS levels. We used DCFH-DA to assess overall ROS production in cells subjected to H/R injury or treatments, providing a broad measure of oxidative stress.

  • Results: Figure 1 A: it is very hard to see important differences in these photomicrographs. The photomicrographs are too bright and the differences should be better explained in the methods part. The reader should be completely introduced what to expect in these photomicrographs. The simple explanation in the results is not enough. This includes photomicrographs in all figures.

Thanks for your review. The original figures were improved by the editor at the moment. We will check the data with high resolution on the website later.

  • TEM - images should have arrows pointing important changes.

Thanks for your advice. The arrows can be detected in the TEM figures .

  • Is flowcytometry protocol anywhere explained (methods)? In Figure 4E, 5E and 6E graphs should be explained - axes need to be explained.

Thanks for your suggestions. Yes, the flowcytometry was mentioned in the Lipid ROS detection in the methods part. In the histogram, each cell is represented along the x-axis by its fluorescence intensity. The Y-axis represents the number of events (cell count) at each fluorescence intensity. So it's the frequency of cells. The graphs and axes in the Figure 4E, 5E and 6E were explained in the revised manuscript.

  • The schematic presentation in Figure 7 is great.Discussion and conclusion are concise and include all necessary information.

We thank the reviewer for their careful scrutiny and happy to hear the evaluation above.

Reviewer 3 Report

Comments and Suggestions for Authors

Lung transplantation (LTx) is associated with severe complications, one of which is ischemia-reperfusion injury (LIRI). Ferroptosis, an iron-dependent form of cell death, has been implicated in the pathogenesis of LIRI. Maresin1 (MaR1) is an endogenous lipid mediator with anti-inflammatory and tissue-protective effects. In this study, the authors investigated whether MaR1 suppresses ferroptosis in LIRI and whether the PKA-Hippo-YAP pathway is involved in this mechanism. The authors established a mouse LTx model and a hypoxic-reoxygenation (H/R) model using human pulmonary microvascular endothelial cells (HPMEC), and demonstrated that MaR1 suppresses ferroptosis by activating the PKA-Hippo-YAP pathway, thereby reducing lung injury.

This study is highly recommended, as it clearly demonstrates that MaR1 suppresses ferroptosis to reduce LIRI. However, the experimental design and discussion are overly focused on ferroptosis, and the following points need further description and explanation.

Major Comments

  1. Pathological evaluations (Figures 1, 2, etc.) are difficult to assess accurately due to low resolution images and low magnification. While it is assumed that pathological scores were calculated based on findings such as alveolar edema, hemorrhage, and inflammatory cell infiltration, it is recommended to explicitly state the scores for each item and to provide representative high magnification images that clearly show these findings for each experimental group.
  2. The ferroptosis-inhibitory effect of MaR1 is demonstrated in the current data; however, MaR1 is known to have general anti-inflammatory and antioxidant effects, and the LIRI-reducing effect may not be specific to ferroptosis inhibition. Therefore, it is necessary to distinguish the effects from those of anti-inflammatory and antioxidant actions or to supplement the experimental data and discussion supporting that ferroptosis is the primary mechanism. In addition, if there are drugs that selectively act on ferroptosis (e.g., ferrostatin-1), it is desirable to include comparisons with such drugs in the description.
  3. Regarding the experiment using sorafenib as a ferroptosis-inducing agent, the discussion suggests that its effects are limited to ferroptosis; however, sorafenib is known to have multifaceted effects, including influencing inflammation, apoptosis, and angiogenesis. Indeed, Martens et al (2017) reported that sorafenib has a tissue protective effect in an inflammatory disease model (DOI: 10.1038/cddis.2017.298). The consistency between these findings and the results of this study should be clearly addressed, at least in the Discussion section.

Minor comments

  1. In this study, in vitro experiments were performed with HPMEC only. However, LIRI involves other cell types such as alveolar epithelial cells and immune cells. The rationale for limiting the study to HPMEC should be discussed, along with considerations regarding the use of other cell types.
  2. Many figures and tables are of low resolution and the text is small, making them difficult to read. Improvements in the overall image quality and layout of figures and tables are required.
  3. With regard to the administration of MaR1, it is noted as a limitation that it was administered as a single dose at a fixed dose. However, the rationale for the choice of this dose as well as the explanations regarding the possibility of changes in efficacy with different doses or administration frequencies are insufficient. Clearer descriptions of these points are required.
  4. Regarding the wet/dry ratio, which is used as an indicator of lung injury, the details of the measurement method are not described in the text. Therefore, it is necessary to clearly specify the methodology in terms of reproducibility.

Author Response

Lung transplantation (LTx) is associated with severe complications, one of which is ischemia-reperfusion injury (LIRI). Ferroptosis, an iron-dependent form of cell death, has been implicated in the pathogenesis of LIRI. Maresin1 (MaR1) is an endogenous lipid mediator with anti-inflammatory and tissue-protective effects. In this study, the authors investigated whether MaR1 suppresses ferroptosis in LIRI and whether the PKA-Hippo-YAP pathway is involved in this mechanism. The authors established a mouse LTx model and a hypoxic-reoxygenation (H/R) model using human pulmonary microvascular endothelial cells (HPMEC), and demonstrated that MaR1 suppresses ferroptosis by activating the PKA-Hippo-YAP pathway, thereby reducing lung injury. This study is highly recommended, as it clearly demonstrates that MaR1 suppresses ferroptosis to reduce LIRI. However, the experimental design and discussion are overly focused on ferroptosis, and the following points need further description and explanation.

Major Comments

  • Pathological evaluations (Figures 1, 2, etc.) are difficult to assess accurately due to low resolution images and low magnification. While it is assumed that pathological scores were calculated based on findings such as alveolar edema, hemorrhage, and inflammatory cell infiltration, it is recommended to explicitly state the scores for each item and to provide representative high magnification images that clearly show these findings for each experimental group.

We are grateful to the reviewer for prompting these critical improvements, which strengthen accuracy and clarity of our findings. This scoring system assesses the following features: â‘  Alveolar congestion: graded based on the extent of congestion visible in alveolar spaces. â‘¡ Hemorrhage: assessed by the presence and severity of bleeding within alveoli and interstitial spaces. â‘¢ Infiltration of inflammatory cells: evaluated by the degree of leukocyte infiltration in alveolar and interstitial regions. â‘£ Thickening of the alveolar walls: based on interstitial edema and cellular proliferation. â‘¤ Hyaline membrane formation: scored according to presence and extent. Each parameter is typically scored on a scale from 0 (normal) to 1 (severe), with the total lung injury score obtained by summing the individual parameter scores, providing an overall assessment of injury severity. The original figures were improved by the editor at the moment. We will check the data with high resolution which can also be magnificated on the website later.

  • The ferroptosis-inhibitory effect of MaR1 is demonstrated in the current data; however, MaR1 is known to have general anti-inflammatory and antioxidant effects, and the LIRI-reducing effect may not be specific to ferroptosis inhibition. Therefore, it is necessary to distinguish the effects from those of anti-inflammatory and antioxidant actions or to supplement the experimental data and discussion supporting that ferroptosis is the primary mechanism. In addition, if there are drugs that selectively act on ferroptosis (e.g., ferrostatin-1), it is desirable to include comparisons with such drugs in the description.

Thanks for your rigorous review. While MaR1 demonstrates protective effects consistent with ferroptosis inhibition, it is important to note that it also possesses anti-inflammatory and antioxidant properties, which may contribute to its overall efficacy. We acknowledge that additional mechanisms, including anti-inflammatory and antioxidant pathways, can initiate ferroptosis and ischemia-reperfusion injury, and they can interact with each other and further exacerbate ischemia-reperfusion injury, which is difficult to completely distinguish. Further studies with drugs that a selectively act on ferroptosis are warranted to delineate these contributions. We add these limitations in the discussion part.

  • Regarding the experiment using sorafenib as a ferroptosis-inducing agent, the discussion suggests that its effects are limited to ferroptosis; however, sorafenib is known to have multifaceted effects, including influencing inflammation, apoptosis, and angiogenesis. Indeed, Martens et al (2017) reported that sorafenib has a tissue protective effect in an inflammatory disease model (DOI: 10.1038/cddis.2017.298). The consistency between these findings and the results of this study should be clearly addressed, at least in the Discussion section.

While sorafenib is widely recognized as a potent inducer of ferroptosis, it is also known to exert additional biological effects, such as modulating inflammation, inducing apoptosis, and inhibiting angiogenesis (Martens et al., 2017). Notably, Martens et al. reported that sorafenib exhibited tissue protective effects in an inflammatory disease model, highlighting its complex pharmacological profile. In our study, sorafenib’s ability to induce ferroptosis appears to contribute significantly to lung injury under hypoxia-reoxygenation conditions; however, these other effects cannot be entirely excluded. The observed protective effects of MaR1 in mitigating sorafenib-induced injury may involve modulation of multiple pathways, including ferroptosis and inflammation. Therefore, the activity of sorafenib in our model aligns with its multifaceted nature, and future investigations dissecting its specific contributions to ferroptosis versus other pathways will enhance our understanding of its role in lung injury. We citated the reference and incorporated this discussion to clarify the broader effects of sorafenib and place our findings within the context of its complex pharmacology.

Minor comments

  • In this study, in vitro experiments were performed with HPMEC only. However, LIRI involves other cell types such as alveolar epithelial cells and immune cells. The rationale for limiting the study to HPMEC should be discussed, along with considerations regarding the use of other cell types.

In this study, we utilized HPMECs to investigate endothelial responses to hypoxia/reoxygenation injury, given that pulmonary endothelial dysfunction is a critical early event in LIRI and plays a central role in modulating subsequent inflammatory and tissue damage responses. The choice of HPMECs was also driven by their availability and well-characterized nature for in vitro vascular studies. However, we acknowledge that LIRI involves complex interactions among multiple cell types, including alveolar epithelial cells, alveolar macrophages, and other immune cells, which contribute to the overall injury process. Future studies should incorporate co-culture systems or primary cell cultures of alveolar epithelial cells and immune cells to better model the cellular interplay and to fully understand MaR1’s effects within the lung microenvironment. Including these additional cell types could provide insights into cell-specific responses and potentially reveal additional mechanisms by which MaR1 confers protection in vivo. We included this explanation in the discussion part to clarify our focus and outline the importance of expanding to other cell types in subsequent research.

  • Many figures and tables are of low resolution and the text is small, making them difficult to read. Improvements in the overall image quality and layout of figures and tables are required.

We sincerely thank the reviewer for their careful scrutiny. The original figures were improved by the editor at the moment. We will check the data with high resolution on the website later.

  • With regard to the administration of MaR1, it is noted as a limitation that it was administered as a single dose at a fixed dose. However, the rationale for the choice of this dose as well as the explanations regarding the possibility of changes in efficacy with different doses or administration frequencies are insufficient. Clearer descriptions of these points are required.

The dose of MaR1 was selected based on previous studies demonstrating effective anti-inflammatory and protective effects in models of organ injury. Similar dosage was employed and reported significant benefits without adverse effects. The administration timing was aimed at providing early intervention during the initial phases of ischemia-reperfusion injury. We acknowledge that the use of a single, fixed dose and administration at a single time point represent limitations, and that the efficacy could be dose-dependent. Future studies should explore various doses and administration frequencies to optimize therapeutic outcomes. Variations in efficacy may occur with higher or lower doses or with repeated administrations, which could enhance or diminish the protective effect depending on pharmacokinetic and pharmacodynamic factors. These considerations will be addressed in subsequent investigations.We incorporated this explanation into the Methods and Discussion sections to clarify our rationale and acknowledge the potential impact of dosing strategies.

  • Regarding the wet/dry ratio, which is used as an indicator of lung injury, the details of the measurement method are not described in the text. Therefore, it is necessary to clearly specify the methodology in terms of reproducibility.

Thank you for your comment. We acknowledge the need to provide detailed methodology for the wet/dry (W/D) ratio measurement to ensure reproducibility. The lung wet/dry (W/D) weight ratio was determined as follows: The lungs were carefully excised and blotted gently with filter paper to remove excess surface blood and moisture. The wet weight was recorded using an analytical balance. The lungs were then dried in an oven at 60°C for 48 hours until a constant weight was achieved. The dry weight was measured with the same balance. The W/D ratio was calculated by dividing the wet weight by the dry weight. We have incorporated this detailed procedure into the Methods section of the manuscript to improve clarity and reproducibility.

Round 2

Reviewer 1 Report

Comments and Suggestions for Authors

The authors have addressed several prior concerns with improved methodological description and expanded discussion. However, a number of important issues remain unresolved or could benefit from further clarification.

Major Issues
1Model Specificity and IRI Attribution: The authors describe standardized procedures to control for confounding factors (e.g., surgical trauma, temperature). However, the study would be strengthened by incorporating or at least discussing the use of ischemia-specific markers—such as HIF-1α or LDH—to delineate IRI from generalized tissue stress. Acknowledging this limitation explicitly would increase transparency and rigor.
2Mechanistic Validation of the PKA–Hippo–YAP Axis: While pharmacological inhibition is informative, the lack of genetic confirmation (e.g., YAP knockdown or PKA activation) limits the causal interpretation of the pathway. Even a brief in vitro validation or a more detailed mechanistic plan for future studies would enhance the credibility of this proposed axis.
3Temporal Resolution of Ferroptosis Dynamics: The justification for the hypoxia-reoxygenation (H/R) timepoints is based on preliminary experiments, but no supporting data are shown. Including representative time-course data (e.g., lipid ROS or ACSL4 expression) as supplemental material would improve methodological clarity and help define the optimal window for ferroptosis activity in this model.
4Ferroptosis Specificity Relative to Other Cell Death Pathways: The authors mention potential crosstalk with apoptosis and necroptosis but provide no comparative data. Even a brief quantification of GPX4 or ACSL4 versus markers of apoptosis (e.g., cleaved caspase-3) or necroptosis (e.g., MLKL) would help determine whether ferroptosis is the dominant pathway under MaR1 regulation in this context.
5Outcome Measures and Translational Perspective: The study focuses on early-stage injury (e.g., within 6–12 hours), but the lack of extended functional or survival endpoints limits translational applicability to clinical lung transplantation. The authors could consider either expanding the observation window in future studies or more clearly justifying this acute-phase focus.
6Contextual Enhancement Through Relevant Literature:The manuscript would benefit from the inclusion of additional studies that have explored the role of MaR1 in ischemia-reperfusion injury and ferroptosis in other organ systems. Recent preclinical investigations have demonstrated that MaR1 exerts protective effects against ferroptosis-mediated injury in both renal and pulmonary inflammatory models, highlighting its broader relevance in systemic ischemic contexts. For example, studies such as:
Int Immunopharmacol. 2024;131:111792. doi:10.1016/j.intimp.2024.111792 and J Pers Med. 2023;13(3):534. doi:10.3390/jpm13030534
have provided supporting evidence for MaR1's immunomodulatory and ferroptosis-inhibiting actions in acute organ injury models. Integrating these findings into the current discussion would enhance scientific continuity, reinforce mechanistic plausibility, and better situate the present study within the evolving MaR1-ferroptosis research landscape.

Minor Suggestions
1Clarify the exact number of biological replicates and independent experiments per group.
2A schematic diagram summarizing the proposed MaR1–PKA–YAP–ferroptosis signaling mechanism would aid reader understanding.
3Consider condensing repetitive result summaries in the Discussion and instead highlight future directions or unresolved mechanistic questions.

Reviewer 2 Report

Comments and Suggestions for Authors

minor: subheading regents....instead of reagents

Author Response

  • minor: subheading regents....instead of reagents

Thank you for pointing out the typo. The subheading has been corrected from "regents" to "reagents" in the revised manuscript. We appreciate your careful review and assistance in improving the clarity of our work.

Reviewer 3 Report

Comments and Suggestions for Authors

The revised manuscript, overall, addresses the previous comments clearly, and the quality of the paper has significantly improved.

However, one major issue remains unresolved. Specifically, the authors have not adequately responded to the comment at the beginning of the “Major Comments” section: “It is recommended to explicitly state the scores for each item”.

Moreover, the authors now state in the revised version: “Each parameter is typically scored on a scale from 0 (normal) to 1 (severe), with the total lung injury score obtained by summing the individual parameter scores, providing an overall assessment of injury severity.” This scoring system appears to be inconsistent with the reference, "Hydrogen inhalation ameliorates ventilator-induced lung injury" (Huang, C. S.; Kawamura, T.; Lee, S.; Tochigi, N.; Shigemura, N.; Buchholz, B. M.; Kloke, J. D.; Billiar, T. R.; Toyoda, Y.; Nakao, A.). Crit Care. 2010; 14(R234). The reference employed a 5-grade scale for each pathological parameter. In practice, it is nearly impossible to accurately assess pathological findings using only two categories, such as "normal" and "severe."

Therefore, the following two revisions are essential:

1) The grading scale should be revised to include more detailed categories (e.g., normal, mild, moderate, severe), and the evaluation should be redone accordingly.

2) As in the reference paper, the scores for each parameter should be presented in a table, allowing readers to understand the pathological context more precisely.

Round 3

Reviewer 2 Report

Comments and Suggestions for Authors

fine

Reviewer 3 Report

Comments and Suggestions for Authors

I appreciate the authors' thorough responses to the comments and concerns raised in the previous review. The revisions satisfactorily address all major issues and significantly improve the manuscript. I recommend accepting the manuscript.